# The Crystal Structure of Engineered Nitroreductase NTR 2.0 and Impact of F70A and F108Y Substitutions on Substrate Specificity

**DOI:** 10.3390/ijms24076633

**Published:** 2023-04-01

**Authors:** Abigail V. Sharrock, Jeff S. Mumm, Gintautas Bagdžiūnas, Narimantas Čėnas, Vickery L. Arcus, David F. Ackerley

**Affiliations:** 1School of Biological Sciences, Victoria University of Wellington, Wellington 6012, New Zealand; abby.sharrock@vuw.ac.nz; 2Wilmer Eye Institute, Johns Hopkins University, Baltimore, MD 21287, USA; jmumm3@jhmi.edu; 3Institute of Biochemistry, University of Vilnius, Saulėtekio 7, LT-10257 Vilnius, Lithuania; gintautas.bagdziunas@gmc.vu.lt (G.B.); narimantas.cenas@bchi.vu.lt (N.Č.); 4Te Aka Mātuatua-School of Science, University of Waikato, Hamilton 3240, New Zealand

**Keywords:** type I nitroreductase, NfsB, crystal structure, prodrug, metronidazole, targeted cellular ablation, tinidazole, CB1954, dinitrotoluene

## Abstract

Bacterial nitroreductase enzymes that convert prodrugs to cytotoxins are valuable tools for creating transgenic targeted ablation models to study cellular function and cell-specific regeneration paradigms. We recently engineered a nitroreductase (“NTR 2.0”) for substantially enhanced reduction of the prodrug metronidazole, which permits faster cell ablation kinetics, cleaner interrogations of cell function, ablation of previously recalcitrant cell types, and extended ablation paradigms useful for modelling chronic diseases. To provide insight into the enhanced enzymatic mechanism of NTR 2.0, we have solved the X-ray crystal structure at 1.85 Angstroms resolution and compared it to the parental enzyme, NfsB from *Vibrio vulnificus*. We additionally present a survey of reductive activity with eight alternative nitroaromatic substrates, to provide access to alternative ablation prodrugs, and explore applications such as remediation of dinitrotoluene pollutants. The predicted binding modes of four key substrates were investigated using molecular modelling.

## 1. Introduction

Nitroaromatic compounds are relatively rare in nature, but many have been synthesised for industrial purposes, e.g., explosives, dyes and pesticides [1]. Nitroreductase enzymes play a critical role in the metabolism of these compounds. Of particular interest are type I (oxygen independent) bacterial nitroreductases, which are flavoenzymes that catalyse the concerted transfer of two electrons from reduced nicotinamide adenine dinucleotide (phosphate) (NAD(P)H) to the nitro group of the substrate. This results in the formation of a reduced nitroso intermediate, which subsequently undergoes further reduction to generate the final hydroxylamine or amine product(s). The resulting changes in electronic charge distribution can greatly alter the reactivity of the substrate molecule, a feature that underpins the classification of many nitroaromatic compounds as prodrugs.

The ability of nitroreductase enzymes to convert nitroaromatic prodrugs into potent cytotoxins has been exploited to develop cancer therapies that target cancer cells while leaving healthy cells unharmed [2]. Nitroreductase genes under transcriptional control of cell type specific promoters have also been used for targeted cellular ablation in transgenic zebrafish, where precise ablation of target cells permits study of the function of specific cell types, or investigation of how discrete cell types are regenerated, including screening for drug candidates to treat diseased states [3]. Recently, the cell death pathways elicited by nitroreductase-mediated ablation have been shown to be relevant to neurodegenerative disease [4] suggesting that this system can also be used as an inducible physiological disease modelling system for conditions linked to selective neuronal cell loss.

For most targeted ablation studies in zebrafish, the nitroreductase NfsB from *Escherichia coli* has been used in partnership with the 5-nitroimidazole prodrug metronidazole [5]. Metronidazole offers substantial advantages for ablation of various zebrafish cell types, owing to its ready availability and affordability, and the cell-entrapped (i.e., ‘nil-bystander’ [2]) nature of its cytotoxic reduction product. It is also generally well tolerated by zebrafish, with concentrations up to 1 mM having no detrimental effect on fish survival or fecundity over 36 days of continuous exposure [6]. However, first-generation ablation systems employing *E. coli* NfsB or a mutant variant thereof [7] frequently required higher metronidazole concentrations to achieve robust ablation of target cells, resulting in non-specific toxicities. Moreover, several cell types have proven resistant to ablation by this approach.

To resolve these issues, we recently engineered an improved nitroreductase, NTR 2.0, that achieves metronidazole-mediated cell-specific ablation at 100-fold lower metronidazole concentrations than required by *E. coli* NfsB [6]. NTR 2.0 was derived from the NfsB orthologue of *Vibrio vulnificus* by introduction of rationally identified F70A and F108Y residue substitutions. NTR 2.0 exhibits faster ablation kinetics and requires lower metronidazole doses, permitting enhanced interrogations of cell function, ablation of previously recalcitrant cell types and novel sustained ablation paradigms that will be useful for modelling of chronic disease states [6]. To better understand the improved enzymatic mechanism of NTR 2.0, we have now crystallised and solved the structure of NTR 2.0 to 1.85 Angstrom resolution. To expand its potential utility, we also provide activity and computational modelling data for NTR 2.0 with a panel of alternative nitroaromatic prodrugs and two prevalent nitroaromatic pollutants.

## 2. Results and Discussion

### 2.1. The Structure of Homodimeric NTR 2.0 and Comparison to Unmodified V. vulnificus NfsB

NTR 2.0 was successfully expressed, purified, and crystallised using a sodium acetate and polyethylene glycol precipitant, resulting in crystals that diffracted to 1.71 Å (Figure 1). The NTR 2.0 crystal belongs to the orthorhombic space group *C* 2 2 2_1_ with unit cell parameters *a* = 68.43, *b* = 137.36, *c* = 107.04, α = 90°, β = 90°, and γ = 90°. The asymmetric unit contained two NTR 2.0 molecules, with a Matthews coefficient of 2.32 Å3/Da and a solvent content of 47% [8]. The structure of NTR 2.0 is a homodimer, with each monomer composed of 14 α-helices and 4 β-sheets (with connecting turns and loops).

As previously reported, NTR 2.0 is a double mutant that exhibits substantially improved metronidazole reductase activity compared to the parental *V. vulnificus* NfsB enzyme [6]. To investigate how amino acid substitutions F70A and F108Y improved activity, we performed a structural comparison with the parental enzyme (PDB code 6CZP). Unfortunately, repeated attempts to capture metronidazole or alternative prodrugs in the active site of the crystal structure proved unsuccessful. It should be noted, however, that nitroreductase enzymes are not thought to undergo significant conformational change upon substrate binding [9,10,11], nor are their amino acid residues directly involved in the chemical reactions associated with substrate oxidation and reduction [12]. Instead, active site amino acid residues act to stabilise the charge on the reduced form of the flavin mononucleotide (FMN) cofactor and form binding pockets that position the substrate, NADPH and FMN at optimal distances for efficient hydride transfer [12,13,14]. As such, the structure of a nitroreductase without a bound substrate likely parallels that adopted when a substrate is bound, and can thus be considered an appropriate model for analysing how the residue configuration in the active site environment may influence substrate binding and catalysis.

In both 6CZP and our NTR 2.0 structure, two monomers interact head-to-tail, resulting in a dimer with a large interface. Each dimer has two active sites, formed at this interface and comprised of residues from both monomers. In both our wild-type and mutant structures, one FMN and one acetate molecule occupy each active site. Inhibition studies have demonstrated that acetate binds only to the oxidised form of a NfsB family nitroreductase and competitively inhibits both substrate and NADPH binding [15]. The tertiary structures of NTR 2.0 and *V. vulnificus* NfsB are compared in Figure 2 and Figure 3.

NTR 2.0 and the parental enzyme share high structural similarity (root-mean-square deviation (RMSD) of 0.32 Å); however, significant differences were noted in the active site (Figure 4).

In the NTR 2.0 structure, mutation of active site residue 108 from phenylalanine to tyrosine introduces hydrogen bonding from the Y108 hydroxyl group to nearby residue E102. This new bond may stabilise the active site, lowering the energy of the transition state during metronidazole catalysis. Additionally, nearby residue F199 is displaced in NTR 2.0 by 1.5 Å, the significance of which is not readily discernible (Figure 5).

Notably, the NTR 2.0 structure also exhibits a widened substrate entrance channel compared to the wild-type, resulting from the displacement of residues F123, F124 and F68 and the removal of the aromatic side chain at residue F70 (Figure 6). It is well established that the tunnels and channels that connect the active site to the external environment can play important roles in catalysis, capable of modulating enzymatic activity, specificity, promiscuity, enantioselectivity and stability [16]. Widening of this channel likely reduces steric hindrance, potentially allowing faster and more efficient passage of NAD(P)H cofactor, reactants and end-products into and out of the active site.

Consurf is a computational tool that aligns an input protein sequence against a set of orthologous sequences from a reference database and assigns a conservation score to each residue [17]. Following assessment of 150 NfsB orthologues, Consurf identified residues 70 and 108 as moderately conserved, each scoring ‘6’ on a 1–9 scale where 9 represents high conservation. Among the 150 orthologues, the residue variety was S/M/Y/A/L/K/E/F at position 70, and I/Y/L at position 108. Thus, both the A70 and Y108 residues introduced into NTR 2.0 can be found naturally in orthologous enzymes.

### 2.2. Surveying NTR 2.0 Activity with Alternative Nitroaromatic Substrates

If a widened substrate access channel/reduced steric hindrance is the main driver behind improved metronidazole activity, we might expect NTR 2.0 to be a generally more efficient enzyme that is improved over the parental enzyme in its activity with a broad range of other nitroaromatic substrates. To investigate this, and to assess the potential of NTR 2.0 for applications beyond metronidazole-mediated ablation, we surveyed the activity of NTR 2.0 with a panel including eight alternative nitroaromatic substrates (structures shown in Figure 7A). The full panel consisted of metronidazole (a 5-nitroimidazole); five alternative nitroaromatic nil-bystander prodrugs (the 2-nitroimidazoles azomycin and RB6145, 5-nitroimidazole tinidazole, and nitrofurans nitrofurazone and nitrofurantoin); CB1954, an anticancer prodrug that induces a bystander effect (i.e., upon activation by a nitroreductase-expressing cell its metabolites can diffuse to and kill neighbouring cells [2]; a feature that could potentially be leveraged for modelling broader injury/trauma paradigms); and 2,4-dinitrotoluene and 2,6-dinitrotoluene, prevalent nitroaromatic pollutants used in munitions, for which efficient nitroreductases are sought for bioremediation applications [18]. Upon nitroreduction, all nine compounds become toxic to *E. coli*, which enabled their activity to be monitored by half maximal inhibitory concentration (IC_50_) growth inhibition assays using nitroreductase-expressing strains.

Relative to unmodified *V. vulnificus* NfsB, NTR 2.0 showed improved activation of 2,6-dinitrotoluene, 2,4-dinitrotoluene, RB6145, tinidazole and metronidazole, and comparable activity with CB1954, nitrofurantoin, nitrofurazone and azomycin (Figure 7B). The preference of each enzyme for 2,4-dinitrotoluene over 2,6-dinitrotoluene was consistent with three NfsA/PnbA family nitroreductases that we previously evaluated in the same background [18]; however, here the IC_50_ values were generally poorer for even the improved NTR 2.0 variant, suggesting that the previous NfsA family nitroreductases hold more promise for bioremediation applications. Of the prodrugs, metronidazole (NTR 2.0 IC_50_ = 15.1 ± 1.5 μM) was most potent, but nitrofurantoin (NTR 2.0 IC_50_ = 29.7 ± 0.1 μM) was comparable, and might provide a suitable alternative for ablation in any scenarios where metronidazole is ineffective. Overall, our data were consistent with the supposition that the widened substrate access channel is generally beneficial to activity with diverse substrates. Other possible impacts of the F70A and F108Y substitutions on active site configuration and substrate binding were explored by molecular modelling.

### 2.3. Molecular Modelling of Prodrug Binding—NTR 2.0 vs. Unmodified V. vulnificus NfsB

We selected four substrates (2,6-dinitrotoluene, CB1954, tinidazole and metronidazole) for docking into the active site of both NTR 2.0 and the parental enzyme (Figure 8). For this, we used a simplified model involving the enzyme active center fragment, an approach that previously proved effective for analysing the specificity of CB1954 reduction by *E. coli* NfsA and NfsB [19]. The isoalloxazine ring of FMN was additionally converted into a reduced form for adequate visualisation of the reaction pathway. This also prevented unproductive orientations of the nitroaromatic compounds toward isoalloxazine, which have been observed in complexes of nitrofurans with oxidised forms of nitroreductases [14,20]. It should be noted that the active centers of 6CZP and NTR 2.0 contain four and five bound H_2_O molecules, respectively (Figure 8A,B), which may be displaced upon the binding of substrate or may interact with it to form H-bonds.

**Metronidazole (MTZ).** The plane-to-plane distance between the imidazole ring of MTZ and isoalloxazine in both enzyme forms is approximately 3.5–3.8 Å (Figure 8C,D). However, the nature of the H-bonding differs between the enzymes: while both form an H-bond between Lys14 and N3 of imidazole, a second H-bond in *V. vulnificus* NfsB is formed between the 2-OH of the ribityl group of FMN and the nitro group, while in NTR 2.0, the second bond is formed between N3-H of isoalloxazine and the ethanol OH group of metronidazole (Figure 8C,D). The calculated distances between the isoalloxazine N5-H and the nitro group oxygen are equal to 4.2 Å (*V. vulnificus* NfsB) and 4.3 Å (NTR 2.0). The calculated E_in_ are −80.9 kJ·mol^−1^ and −84.1 kJ·mol^−1^, respectively.**Tinidazole (TNZ).** As with MTZ, differences in H-bond formation are observed: in unmodified *V. vulnificus* NfsB, the H-bonds are formed between the tinidazole sulfonyl group and Lys14 and Lys74, whereas in NTR 2.0, Lys14 interacts with N3 of the imidazole ring, and the sulfonyl interacts with N3-H of isoalloxazine (Figure 8E,F). The distances between the oxygen of the nitro group and the isoalloxazine N5-H are equal to 4.4 Å (*V. vulnificus* NfsB) and 4.7 Å (NTR 2.0), respectively. The calculated E_in_ are −32.7 kJ·mol^−1^ and −80.9 kJ·mol^−1^, respectively.**2,6-Dinitrotoluene (2,6-DNT).** When bound to both enzymes forms, the compound is nearly parallel with respect to the isoalloxazine ring of FMNH_2_, but is not involved in the π–π interaction with the isoalloxazine or Phe70 of *V. vulnificus* NfsB (Figure 8G,H). The nitro groups of 2,6-DNT form H-bonds with Lys14 and Lys74, but do not interact with other amino acids. The calculated distances between the isoalloxazine N5–8 and the nitro group oxygen are equal to 6.7 Å (parent enzyme) and 6.2 Å (NTR 2.0), whereas E_in_ are equal to −50.6 kJ·mol^−1^ and −54.3 kJ·mol^−1^, respectively.**CB1954.** The aziridine ring of this compound interacts with residue 70 in both enzymes, and despite minor differences in orientation it is the 4-nitro group rather than the 2-nitro group that is oriented toward the isoalloxazine in each case (Figure 8I,J). The distances between the oxygen of the reducible 4-nitro group of CB1954 and the H-N5 of isoalloxazine are equal to 5.5 Å (*V. vulnificus* NfsB) and 5.1 Å (NTR 2.0), whereas E_in_ are equal to −75.1 kJ·mol^−1^ and −56.4 kJ·mol^−1^, respectively. The 4-nitro group forms H-bonds with Lys14 and Lys74.

### 2.4. Analysis of CB1954 Reduction Products

The ability of activated CB1954 metabolites to cause a local zone of cell killing via diffusion from nitroreductase-expressing to neighbouring cells (i.e., the bystander effect; [2]) offers prospects to model broader injury paradigms such as traumatic brain injury. In general, nitroreduction of CB1954 can occur at either the 2-nitro or the 4-nitro functional group, but not both, to generate derivatives that elicit cytotoxicity through distinct mechanisms [21,22,23]. Our substrate modelling suggests that the 4-nitro group of CB1954 is more likely to be reduced than the 2-nitro group, by either NTR 2.0 or the *V. vulnificus* NfsB parental enzyme. Reduction at the 4-nitro position yields a more potent cytotoxin capable of inter-strand DNA cross-linking via an N-acetoxy intermediate [24], whilst reduction at the 2-nitro position produces cytotoxic derivatives that that form less-toxic DNA monoadducts but exhibit a substantially higher bystander effect, with increased diffusion ranges having been observed in 3D cell cultures [21] and tumour spheroid models [25]. It has been suggested that exclusive reduction of CB1954 at the 4-nitro position is preferable for gene therapy applications to maximise cytotoxicity [26], but potential advantages of the heightened bystander effect resulting from reduction at the 2-nitro position have also been noted [27]. At this stage it is unclear to us which metabolites might be preferable for cell ablation models, but we considered it important to define the metabolites produced by NTR 2.0 to inform the field.

To investigate the reductive preference of NTR 2.0, the relative levels of 2- vs. 4-hydroxylamine end-products generated were measured by reverse-phase high-performance liquid chromatography (HPLC) (Figure 9). In addition, the CB1954 reduction end-products of the parental *V. vulnificus* NfsB enzyme, NfsA from *E. coli* and YfkO from *Bacillus subtilis* were examined (the latter two known to exclusively generate the 2-hydroxylamine and 4-hydroxylamine end-products, respectively [27,28,29], and thus serving as internal positive controls for each metabolite). NTR 2.0 and the parental enzyme reduced CB1954 nearly exclusively at the 4-nitro position to generate the 4-hydroxylamine end-product. We note that this contrasts with data one of us (DFA) previously reported, where a 25:75 end-product ratio of the 2-hydroxylamine:4-hydroxylamine was observed for *V. vulnificus* NfsB [28]. This may be due to the slightly different variants of *V. vulnificus* NfsB tested in each study (with NTR 2.0 and its parental NfsB sequence derived from *V. vulnificus* CMCP6, while the authors of [28] used the NfsB protein from *V. vulnificus* ATCC 27562; the two native sequences sharing 98% sequence identity).

## 3. Materials and Methods

### 3.1. Bacterial Strains, Media and Growth Conditions

The *E. coli* strains 7NT (which bears scarless in-frame deletions of seven candidate NTR genes (*nfsA*, *nfsB*, *azoR*, *nemA*, *yieF*, *ycaK*, and *mdaB*) and the efflux pump gene *tolC* [30]) and BL21 (Novagen) were grown in LB medium at 37 °C with agitation (200 rpm) or on LB agar plates at 37 °C (unless otherwise stated).

### 3.2. Cloning, Expression and Purification of NTR 2.0

The gene encoding NTR 2.0 was ordered pre-synthesised from Twist Bioscience (San Francisco, CA, USA). The gene fragment was cloned into pET28a(+) (Novagen) which expresses a His_6_ tag at the C-terminus. The expression construct was used to transform *E. coli* BL21 cells and grown in LB medium containing 50 µg·mL^−1^ kanamycin at 37 °C. After induction with 0.5 mM IPTG, the culture was incubated for a further 24 h at 18 °C. The culture was harvested by centrifugation at 2600× *g* for 20 min. NTR 2.0 was purified using nickel affinity chromatography where bound protein was eluted over a two-step wash and elute method, involving an initial wash at 60 mM imidazole to remove loosely bound non-target proteins and a second wash at 1 M imidazole to elute NTR 2.0. For further purification, size exclusion chromatography was performed using a Superdex S200 10/300 GL column (GE Healthcare, Pollards Wood, UK) connected to an ÄKTA BasicTM FPLC system and a 40 mM Tris-HCl pH 7.0, 200 mM NaCl buffer. Fractions containing the target protein were assessed for purity by SDS-PAGE analysis and quantified by their 280 nm absorbance trace.

### 3.3. Crystallisation

For crystallisation, NTR 2.0 was concentrated to 10 mg·mL^−1^ in a buffer consisting of 40 mM Tris-HCl pH 7.0 and 200 mM NaCl. Initial protein crystallisation trials employed large-scale crystallisation screens covering 288 conditions (Hampton Research screening trays Index, PEGRx1 and Crystal Screen) and used the sitting drop vapour diffusion method in 96-well plates at 291 K (18 °C). NTR 2.0 protein solution (100 nL) was mixed with precipitant solution (100 nL), and equilibrated beside a 100 µL reservoir of precipitant solution. Further crystallisation screens employed hanging-drop vapour diffusion in 24-well VDX plates (Hampton Research, Aliso Viejo, CA, USA) at 291 K. Crystals of NTR 2.0 formed within two weeks by mixing 1 µL of 10 mg/mL protein solution (40 mM Tris-HCl pH 7.0 and 200 mM NaCl) with 1 µL of precipitant solution (0.1 M sodium acetate trihydrate pH 4.5 and 32% (*v*/*v*) polyethylene glycol 300) and by equilibrating over a 500 µL reservoir of precipitant solution. Crystals were orthorhombic and yellow in colour, indicating the presence of the bound FMN cofactor in the oxidised state. Crystals were flash-frozen in liquid nitrogen prior to data collection.

### 3.4. X-ray Data Collection

Due to the high polyethylene glycol concentration in this crystallisation condition, no additional cryo-protectant was required prior to crystal freezing/testing. For data collection, crystals were mounted in a stream of N_2_ gas at 100 K. Data were collected at the macromolecular crystallography beamline (MX1) at the Australian Synchrotron, Melbourne, Australia using an ADSC Quantum 210r detector at a wavelength of 0.9537 Å using 0.5° oscillation per image with a crystal-to-detector distance of 250 mm. A data set was collected to 1.71 Å resolution from a single crystal. All data processing was conducted using software found within CCP4 program suite 6.4.0 (Collaborative Computational Project, Number 4, 1994) [31]. The data set was indexed and integrated using iMosflm 7.0.9 [32]. The integrated and combined reflections were scaled and merged using SCALA [33]. Data were trimmed to a resolution such that the outer shell *R*_merge_ was <0.9 and the final structure was solved to 1.85 Å using molecular replacement with the starting model PDB code 1DS7. The data collection statistics are summarised in Table 1.

### 3.5. Structure Solution and Refinement

The NTR 2.0 structure was solved to 1.85 Å using molecular replacement using the structure of NfsB from *E. coli* (PDB code 1DS7, 60.4% identity) as a model. Molecular replacement was carried out using Phaser [34] in the CCP4 suite (Collaborative Computational Project, Number 4, 1994). Cycles of manual model building using COOT 0.7 [35] followed by refinement in Refmac5 (CCP4 suite). Coordinate files and structure factors for the full structure have been deposited in the Protein Data Bank with PDB code 7UWT. The refinement statistics are included in Table 1.

### 3.6. Bacterial Cytotoxicity Assays

The gene encoding NTR 2.0 was PCR amplified and cloned into the NdeI and SalI sites of the expression plasmid pUCX (Addgene no. 60681). To assess NTR 2.0 properties in a minimal nitroreductase background, the *E. coli* 7NT strain was used, which bears scarless in-frame deletions of seven candidate NTR genes (nfsA, nfsB, azoR, nemA, yieF, ycaK, and mdaB) and the efflux pump gene tolC [30]. *E. coli* 7NT pUCX:NTR2.0 cells were used to inoculate wells of a sterile 96-well microplate containing 200 µL LB supplemented with 100 µg·mL^−1^ ampicillin and 0.2% (*w*/*v*) glucose. Cultures were incubated overnight at 30 °C, 200 r.p.m. The following morning, cultures were diluted 20-fold in induction medium (LB, 100 µg·mL^−1^ ampicillin, 50 μM IPTG). Cultures were incubated at 30 °C, 200 r.p.m. for 2.5 hours. Culture aliquots (30 µL) were added to sterile 384-well microplates containing 30 µL induction medium with or without twice the desired final prodrug concentration. For dose–response tests, each culture was exposed, in duplicate, to increasing titrations of prodrug (15 conditions containing two- or 1.5-fold stepwise increases) and one induction medium-only control. The medium was supplemented with DMSO as appropriate, ensuring that the final DMSO concentration remained <4%. OD600 readings were recorded using an EnSpire 2300 Multilabel Reader (PerkinElmer), and initial OD600 values were within the range of 0.12–0.18. Cultures were incubated at 30 °C, 200 r.p.m. for a further 4 hours, after which OD600 readings were recorded once more. The increase in OD600 of challenged and unchallenged wells for each strain was compared and used to calculate percentage growth at each prodrug concentration. GraphPad Prism 8.0 was used to calculate prodrug IC_50_ values using a dose–response inhibition four-parameter variable slope equation.

### 3.7. HPLC Identification of CB1954 Nitroreduction Products

Reaction mixes of 100 µL comprising 10 mM Tris-HCl pH 7.0, 1 mM NADPH and 200 µM CB1954 were initiated by addition of 1.75 µM purified enzyme. Reactions were incubated for 25 min at room temperature before being stopped by addition of one volume ice-cold 100% methanol. Samples were transferred to −80 °C for at least 1 h to precipitate proteins after which samples were centrifuged for 10 min at 12,000× g, 4 °C. The supernatant was decanted and diluted 1:20 in 45 mM ammonium formate buffer, 2.5% (*v*/*v*) methanol (pH 6.5). A 100 µL volume of each sample was analysed by reverse phase-HPLC employing an Agilent 1200 series system with an Ascents C8 3 µm 150 × 4.6 mm column (Sigma-Aldrich, St. Louis, MO, USA). The mobile phase used for HPLC analysis was 45 mM ammonium formate buffer (pH 6.5) as aqueous and 80% acetonitrile as organic. The HPLC run parameters consisted of 4 min at 5% organic, a linear increase to 50% organic from 4 to 19 min and a further gradient increase to 70% organic from 19 to 21 min. The flow rate was maintained at 1.5 mL·min^−1^ throughout, and the eluate was monitored at 262 nm. Elution profiles from each nitroreductase-CB1954 reaction were compared against control reactions for *B. subtilis* YfkO or *E. coli* NfsA, which have consistently been shown to exclusively reduce CB1954 at the 4- or 2-nitro positions, respectively [27,28,29].

### 3.8. Theoretical Modelling

For computer modelling, we used the crystal structures of *V. vulnificus* NfsB (PDB code 6CZP) and NTR 2.0 (PDB code 7UWT). A simplified modelling approach was used involving the enzyme active center fragment containing the amino acid residues at a radius of 11 Å around the N5 atom of isoalloxazine ring of FMN [19]. FMN was converted into the reduced form (FMNH_2_). All the atoms heavier than hydrogen were frozen, except side chains of amino acids, which can form H-bonds with the substrates (Ser40, Glu165, Lys14, Lys74, Phe124, Phe70, Phe123). The H_2_O molecules present in the starting structures were left to move freely in the active center. Previously, we have shown that water molecules are important for prediction of supramolecular host-guest structure [36]. The corresponding substrate was placed in the active center, and the obtained structures were relaxed and optimised using a molecular mechanics force field (MMFF) method. Dispersion-correcting potential approach of density function theory (B3LYP-D3) and 6–31(d,p) basis set in vacuum was applied to calculate the single point heat of formation energies of these obtained structures. The energies of substrate binding (E_in_) were calculated using Equation (1):E_in_ = E(pocket-S) − E(pocket) − E(S)(1)
where E(pocket-S), E(pocket), and E(S) are the heats of formation of the pocket-substrate, the free pocket, and free substrate, respectively. All the computations were performed using the Spartan’18 for Windows Version 1.3.0. (Wavefunction, Inc., Irvine, CA, USA).

## 4. Conclusions

In this study, we performed structural and activity analyses of NTR 2.0 to decipher its substrate selectivity with a variety of nitroaromatic prodrugs. Relative to the parental *V. vulnificus* NfsB enzyme, NTR 2.0 was not impaired with any of the compounds tested and exhibited improved activity with 2,6-dinitrotoluene, 2,4-dinitrotoluene, RB6145, tinidazole and metronidazole. Reduction of CB1954 by NTR 2.0 yielded a 4-hydroxylamine product that is likely to generate a small local bystander effect, which could potentially be leveraged for modelling broader injury/trauma paradigms in transgenic animal models. The X-ray crystal structure of NTR 2.0 revealed a widened substrate access channel compared to the wild-type structure, which may contribute to the generally increased activity. The data from computer simulations show that key nitroaromatic substrates bind in the active sites of both NTR 2.0 and the parental enzyme in productive orientations, with the calculated distances between the N5-H of isoalloxazine and the oxygen of the nitro group varying between 6.7 and 4.2 Å. However, because of the significant data scattering, these data may be considered as approximate only. The inferred energies of nitroaromatic substrate binding are comparable to that of binding of nitrofurantoin to reduced NfsA, −91.6 kJ·mol^-1^, calculated in a similar way [20]. Most likely, the F70A substitution affects the reactivity of nitroaromatic compounds in a complex and distant way, by an increased accessibility of isoalloxazine, and through changes in the system of H-bonds and hydrophobic interactions. On the other hand, in the case of all compounds, an important role of Lys14 and sometimes Lys74 in forming H-bonds with nitro or other proton accepting groups of nitroaromatics was predicted.

Together, these structural and activity analyses shed light on the likely molecular mechanisms underpinning the improved activation of metronidazole, as well as additional substrates that may enhance the versatility of NTR 2.0 for a range of applications.

## Figures and Tables

**Figure 1 ijms-24-06633-f001:**
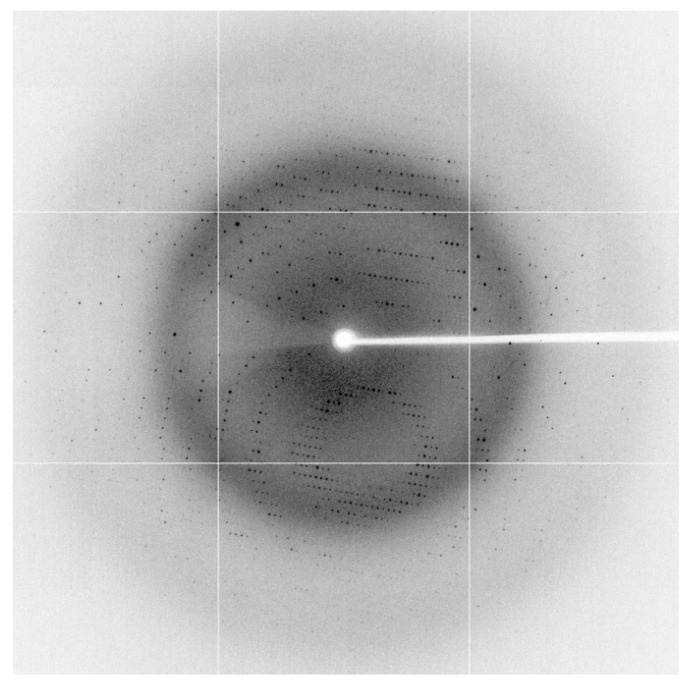
Representative X-ray diffraction image from NTR 2.0 crystal. The edge of the detector corresponds to a resolution of 1.71 Å.

**Figure 2 ijms-24-06633-f002:**
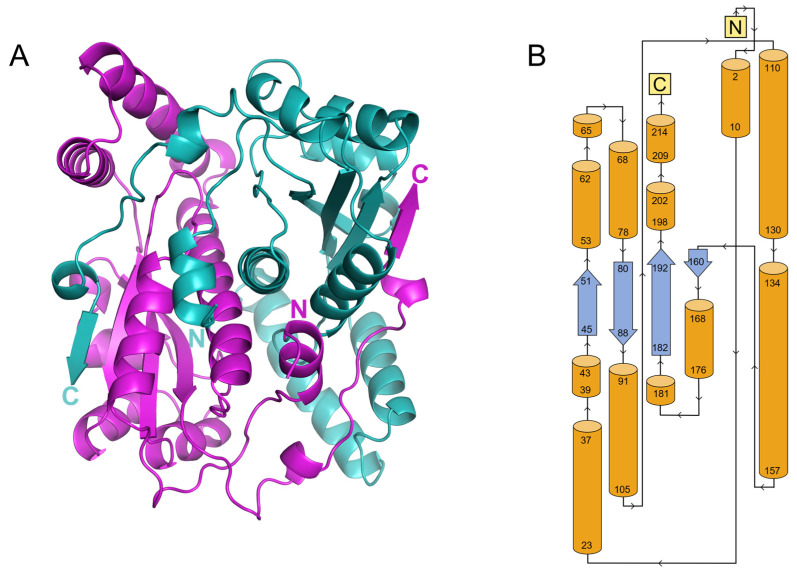
Cartoon structural representation of NTR 2.0. (**A**) Side view of NTR 2.0 homodimer where individual monomers are coloured cyan or magenta and N and C termini are labelled. (**B**) Topology structure diagram of NTR 2.0 monomer. Numbers represent residue positions. Orange cylinders represent α-helices and blue arrows represent β-strands.

**Figure 3 ijms-24-06633-f003:**
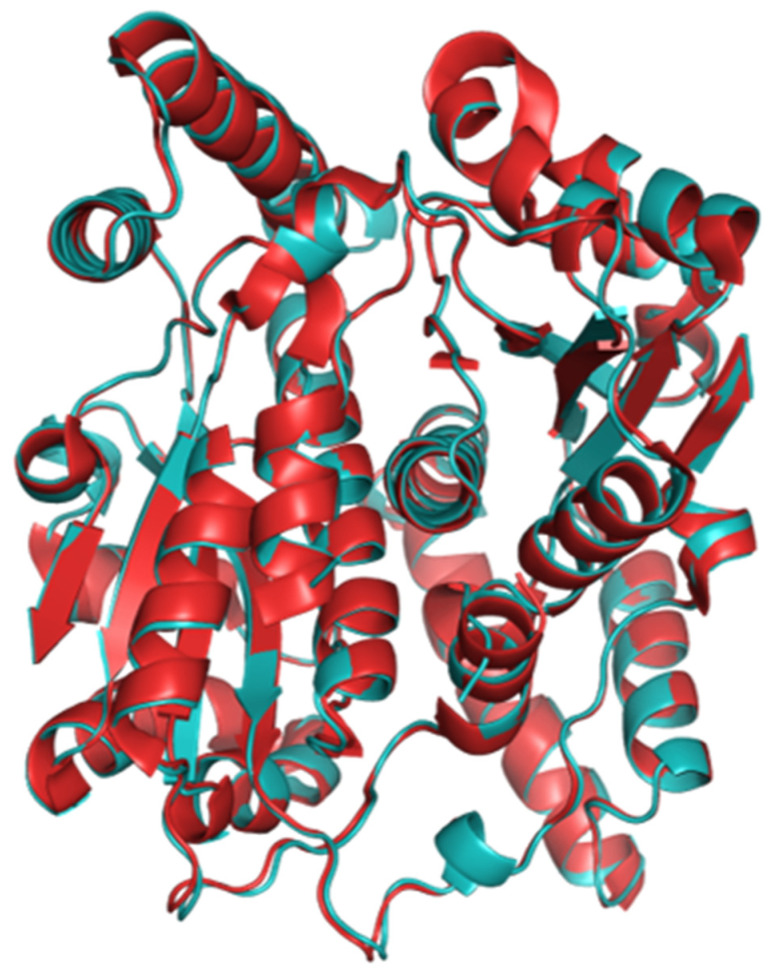
Cartoon overlay of the structures of NTR 2.0 and parental *V. vulnificus* NfsB. The parental enzyme is depicted in red and NTR 2.0 in cyan.

**Figure 4 ijms-24-06633-f004:**
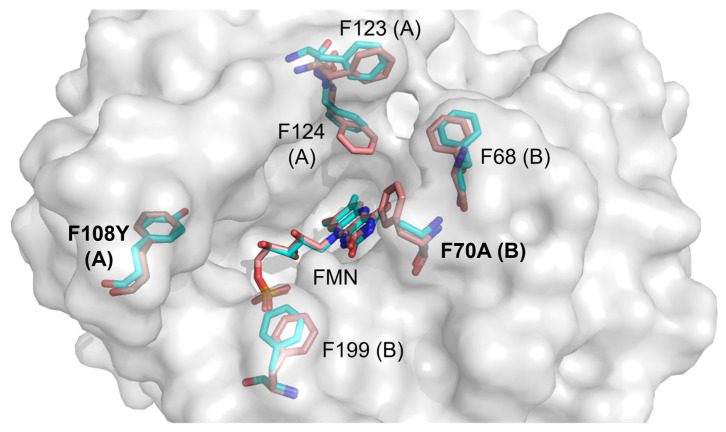
Active site overlay of NTR 2.0 and the parental enzyme. The surface represents the NTR 2.0 structure. Wild-type (red) and NTR 2.0 mutant (cyan) residues exhibiting significant differences and FMN molecules are shown as sticks. Chain identity (A or B) is identified in parentheses after each numbered residue.

**Figure 5 ijms-24-06633-f005:**
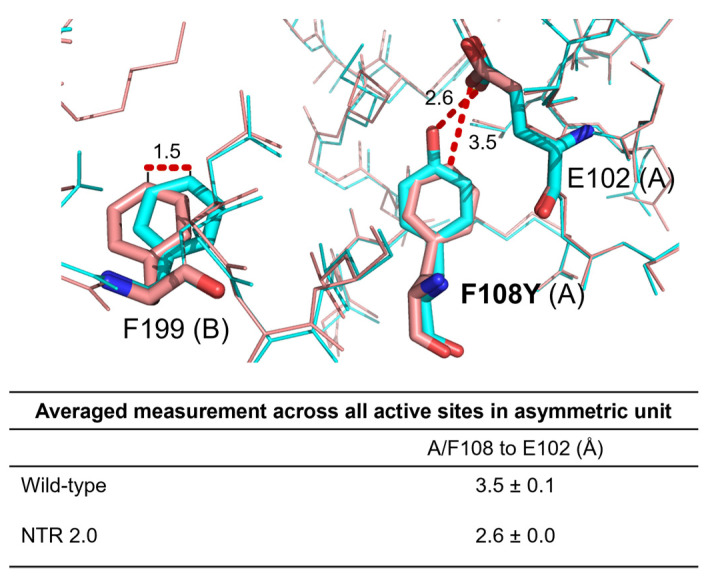
Residue 108 interactions in the active sites of NTR 2.0 and the parental enzyme. Residues at positions 102, 108 and 199 are shown as sticks and coloured red for unmodified *V. vulnificus* NfsB or cyan for NTR 2.0. Distance measurements are in Å. The distance between residue 102 and 108 was measured in PyMOL for each active site present in the asymmetric unit of each structure and the average measurement ± SD was calculated. Chain identity (A or B) is identified in parentheses after each numbered residue.

**Figure 6 ijms-24-06633-f006:**
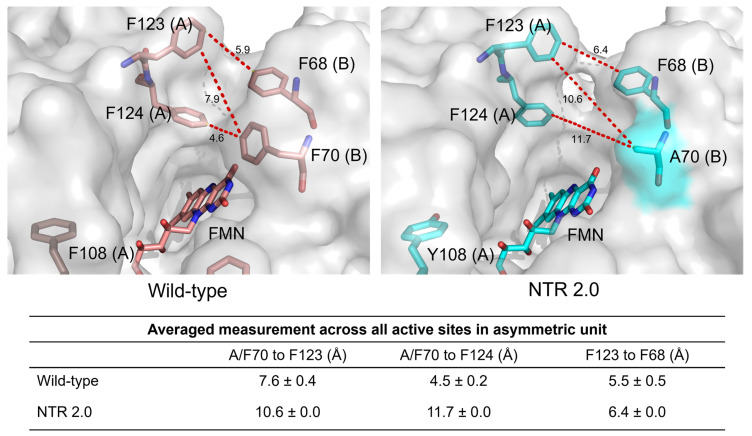
Active site entrance channel width in NTR 2.0 and unmodified *V. vulnificus* NfsB. Residues at positions 68, 70, 108, 123 and 124 and FMN are shown as sticks and measurements are in Å. Distances between residues 123 and 70, 124 and 70, and 123 and 68 were measured in PyMOL for each active site in the asymmetric unit of each structure and the average measurement ± SD was calculated. In each structure, measurement was taken from the F124 atom closest to F70. Chain identity (A or B) is identified in parentheses after each numbered residue.

**Figure 7 ijms-24-06633-f007:**
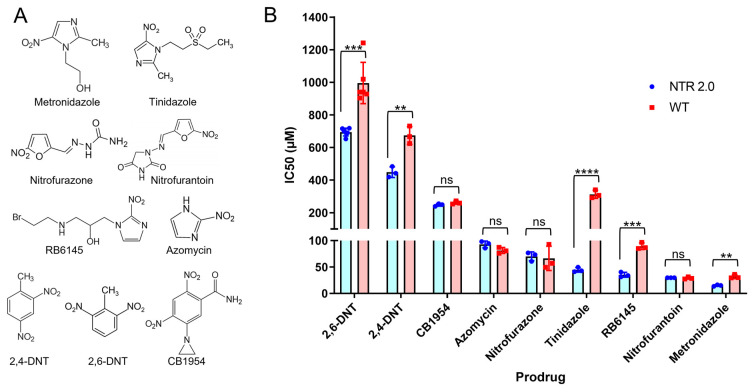
Activity of NTR 2.0 with a panel of nitroaromatic prodrugs. (**A**) Structures of the nitroaromatic prodrugs tested in this work. (**B**) IC_50_ values for *E. coli* cells expressing NTR 2.0 or the parental *V. vulnificus* NfsB enzyme (wild-type; WT) following challenge with each prodrug. Notation above bars denotes statistical significance between strains: ** = *p* ≤ 0.01, *** = *p* ≤ 0.001 and **** = *p* ≤ 0.0001 and ns = *p* > 0.05 (unpaired t-tests). Raw data from the IC_50_ analyses are available as Appendix A.

**Figure 8 ijms-24-06633-f008:**
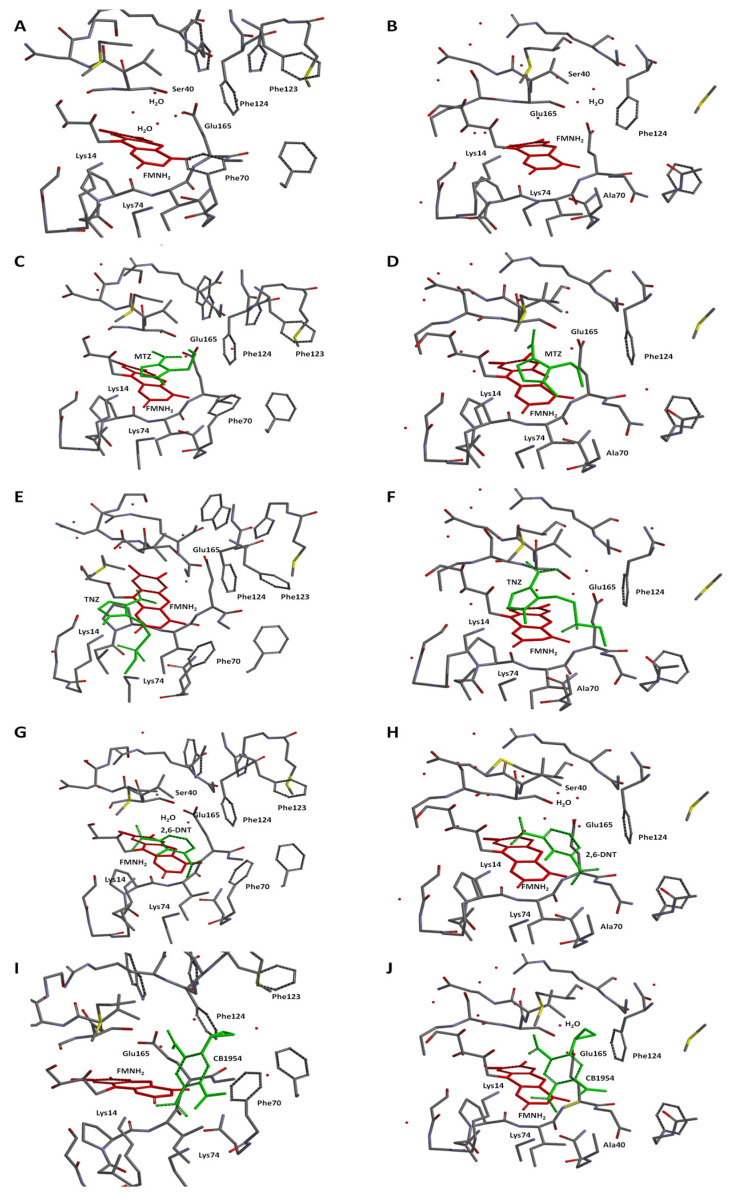
Optimised structures of the active center fragments of *V. vulnificus* NfsB (Left panels; (**A**,**C**,**E**,**G**,**I**) and NTR 2.0 (Right panels), (**B**,**D**,**F**,**H**,**J**) without (**A**,**B**) or with (**C**–**J**) bound substrates. Substrates: metronidazole (MTZ; (**C**,**D**)); tinidazole (TNZ; (**E**,**F**)); 2,6-dinitrotoluene (2,6-DNT; (**G**,**H**)); and CB1954 (**I**,**J**). The isoalloxazine ring of FMNH_2_ is marked in red, nitroaromatic substrates marked in green, water molecules marked by red dots. Hydrogen atoms were removed for clarity.

**Figure 9 ijms-24-06633-f009:**
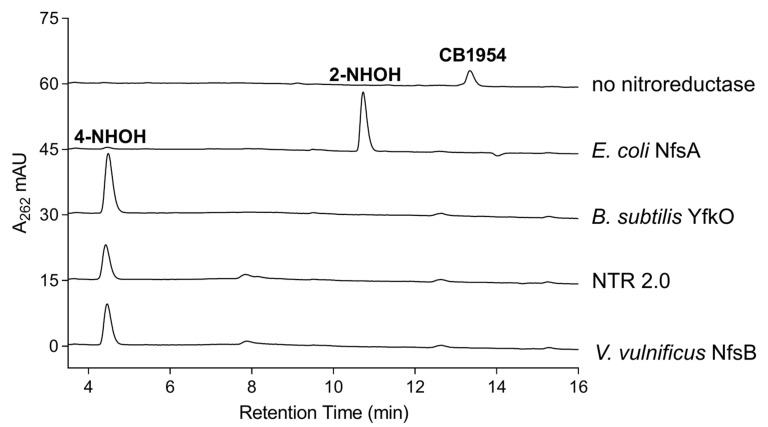
HPLC analysis of reaction products of CB1954 reduction by purified nitroreductases. Purified His_6_-tagged nitroreductase was incubated with CB1954 (100 µM) and NADPH (1 mM) in 10 mM Tris pH 7.0 for 25 min at room temperature prior to chromatographic separation of reaction end-products. Column eluates were monitored at 262 nm. Spectral peaks corresponding to the 4-hydroxylamine (4-NHOH) and 2-hydroxylamine (2-NHOH) end-products, or non-reduced CB1954, were determined by comparison with control traces for *B. subtilis* YfkO (known to exclusively reduce CB1954 at the 4-NO_2_ position [28]), *E. coli* NfsA (known to exclusively reduce CB1954 at the 2-NO_2_ position [28]) or a “no nitroreductase” negative control.

**Table 1 ijms-24-06633-t001:** Data collection and refinement statistics for NTR 2.0.

Wavelength (Å)	0.9537
Space group	*C* 2 2 2_1_
Unit-cell parameters (Å, °)	*a* = 68.43, *b* = 137.36, *c* = 107.04, α/β/γ = 90/90/90
Resolution range (Å)	68.68–1.85 (1.89–1.85)
Measured reflections	624,970
Unique reflections	43,344
Multiplicity	14.4
Temperature (K)	100
Matthews coefficient (Å^3^ Da^−1^)	2.32
Solvent content (%)	47
No. of molecules in ASU	2
Completeness (%)	99.9 (99.7)
Mean *I*/σ(*I*)	18.3 (3.7)
*R*_merge_^†^ (%)	12 (79.2)
CC(1/2)	0.999
Wilson *B* factor (Å^2^)	10.275
R_work_	0.165
R_free_	0.196
Protein atoms	3342
Other ions/molecules	7
Number of waters	184
B factors (proteins)	18.5 (Chain A)/19.8 (Chain B)
B factors (water)	23.9
**RMSD**	
Bond angles (°)	1.6771
Bond lengths (Å)	0.0120

Values in parentheses are for the highest resolution shell. ^†^
*R*_merge_ = Σ_hkl_ Σ_i_|Ii(hkl) − <I(hkl)> |/Σ_hkl_ Σ_i_Ii(hkl), where Ii(hkl) is the observed intensity of an individual reflection and <I(hkl)> is the mean intensity of that reflection.

## Data Availability

Structural data associated with this study are available from the Protein Data Bank (PDB code 6CZP). All other data are available upon request.

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
