# Peer review of "The Crystal Structure of Engineered Nitroreductase NTR 2.0 and Impact of F70A and F108Y Substitutions on Substrate Specificity"

_ijms, 2023, doi:10.3390/ijms24076633_

Round 1
Reviewer 1 Report
Attached as a separate file

Author Response
Reviewer One
The following comments needs to be addressed before the manuscript is accepted for publication.
- In Figure 2A, the colouring pattern is confusing. Use only one colour per monomer. Shadows are visible in the figure. Use shadows off option when using pymol. The shadows are visible in other figures generated by Pymol as well. Kindly turn off shadows when making the figures for publication purpose. It will be good to split this figure to have a separate figure with the current mutant structure, each monomer in separate colour and topology diagram showing only two colours (one for sheet and other for helix) so that the readers can follow easily. Also clearly mark the N and C terminals in the cartoon figure.
We thank the reviewer for these suggestions, and have split the figure and changed the colour scheme from rainbow to monochromatic for each monomer (cyan and magenta). We also recreated the figure with the pymol shadow option turned off – we note that there is still a limited amount of shadow applied to create the three-dimension representation, but this is far less than before. Likewise, the other figures also retain a minimal level of shadowing as they would become meaningless without this three-dimensionality.
- Suggest combining Table 1 and Table 2 as generally reported in other studies. Also please change R factor to R work in Table 2.
These suggestions have also been adopted.
- If possible, change the background colour (to white) for figure 7, it is difficult to distinguish between the black background and light dark stick representations for the residues. Otherwise use a better colour combination so that the residues can be easily visible from the background.
We agree, and have adopted this reviewer’s first suggestion, changing the background colour to white.
- Also kindly explain the large variations in the raw data between the three replicates for the IC50 calculations. What might be the contributing factors?
These assays are true independent biological repeats, i.e. prepared from independently sourced cultures over multiple days. In our experience these error bars are in fact remarkably small for biological repeats of IC50 datasets, and we do not consider it necessary to explain them.
Minor comments.
Type, line 92, acto---- change to act
This typographical error was also noted by Reviewer 2, and has been corrected.
Reviewer 2 Report
The authors successfully demonstrate the molecular mechanism of changes in substrate specificity and improved activity of NTR 2.0 in comparison to a wild type (WT) NfsB enzyme from V. vulnifucus.
Overall, I am happy with this body of work and believe it is suitable for publication in this journal, proving the application of the minor changes listed below. These alterations are primarily aimed at improving the readabilty and no additiona experiments should be required.
Line 92 replace "acto" with "act to"
Line 100 states protein is a dimer. It would be useful to have a native-PAGE gel or SEC trace in supplementary info to confirm this isn't an artefact of crystallization.
Figure 2(A) shows the head to tail dimer arrangement of NTR2.0. It is difficult to distinguish between the 2 chains and the figure would perhaps be clearer if only one colour is used per chain with N and C termini clearly labelled.
Figures 3, 4 and 5: In the text (line 100-101) it states that both chains contribute residues to the active sites. In these figures, please could you identify the chains by adding and "A" or "B" to residues as appropriate. In addition, the yellow dashed lines indicating distances are hard to see clearly, I suggest using a different colour to make them stand out from the figure more.
Figure 6 (B): Shows the IC50's of various prodrugs using either WT enzyme or NTR2.0. It would be useful to do some statistics on the data to determine which changes are significant (e.g. T test).
Author Response
Reviewer Two
Line 92 replace "acto" with "act to"
As noted above, this has been done
Line 100 states protein is a dimer. It would be useful to have a native-PAGE gel or SEC trace in supplementary info to confirm this isn't an artefact of crystallization.
We do not consider this necessary – it is well-established that NfsB family nitroreductases are dimeric (for a global analysis see Akiva et al, Evolutionary and molecular foundations of multiple contemporary functions of the nitroreductase superfamily, Proc Natl Acad Sci USA 114 (45) E9549-E9558) and more specifically, the dimeric nature of the parental Vibrio vulnificus NfsB enzyme was established by the previous PDB structure. Moreover, as each active site is comprised of residues contributed by each individual monomer, these enzymes could not function as isolated monomers.
Figure 2(A) shows the head to tail dimer arrangement of NTR2.0. It is difficult to distinguish between the 2 chains and the figure would perhaps be clearer if only one colour is used per chain with N and C termini clearly labelled.
We thank both reviewers for this helpful suggestion, which we have adopted.
Figures 3, 4 and 5: In the text (line 100-101) it states that both chains contribute residues to the active sites. In these figures, please could you identify the chains by adding and "A" or "B" to residues as appropriate. In addition, the yellow dashed lines indicating distances are hard to see clearly, I suggest using a different colour to make them stand out from the figure more.
This is another very helpful suggestion, which we have adopted.
Figure 6 (B): Shows the IC50's of various prodrugs using either WT enzyme or NTR2.0. It would be useful to do some statistics on the data to determine which changes are significant (e.g. T test).
This has been done, and significant relationships are now indicated in the figure (now Fig 7B).
Round 2
Reviewer 1 Report
The authors have addressed all my comments and I recommend this manuscript for publication.